# Research on Chilling Requirements and Physiological Mechanisms of *Prunus mume*

Yuhan Zhang, Kaifeng Ma and Qingwei Li *

School of Landscape Architecture, Beijing Forestry University, Beijing 100083, China; yuhan199496@163.com (Y.Z.); makaifeng@bjfu.edu.cn (K.M.)
* Correspondence: lqw6809@bjfu.edu.cn

**Abstract:** Low temperature plays a vital role in the growth and development of woody plants. In this research, based on the ability of artificial low temperatures to break dormancy, the Utah model was used to determine the chilling requirements (CR) of four early flowering *Prunus mume* Sieb. et Zucc. (Rosaceae) cultivars, which are widely found in the Henan area of China. In addition, changes in the carbohydrates, antioxidant enzyme activities, and endogenous hormone contents of the flower buds (FBs) of the above-mentioned *P. mume* cultivars were measured during the low-temperature storage process, and the physiological changes of the four cultivars during the low-temperature induction period were explored. The main research conclusions are as follows: (1) the CRs of 'Gulihong', 'Nanjing gongfen', 'Zaoyudie', and 'Zaohualve' were 408CU, 396CU, 372CU, and 348CU, respectively. All the *P. mume* cultivars belonged to cultivars with a low chilling demand. They also bloomed 4 months earlier, and (2) during the process of releasing dormancy at low temperatures, the contents of soluble sugar (SS) in the osmoregulation system of the four *Prunus mume* cultivars showed an upward trend, while the content of starch (ST) basically showed an opposite trend to the increase in chilling. The superoxide dismutase (SOD) activity in the FBs of each cultivar gradually decreased with the increase in cold and remained at a low level, while the peroxidase (POD) activity showed an opposite trend, and the dynamic changes of the catalase (CAT) activity generally showed a trend of first increasing and then decreasing. The content of abscisic acid (ABA) showed a trend of increasing first and then decreasing as a whole. The changing trend of gibberellin (GA$_3$) content was similar to that of ABA. In addition, it was found that before dormancy was released, the SS and Pro contents of cultivars with lower CRs and early FB germination were significantly higher than those of other cultivars with higher CRs and the contents of ST and SP were lower. The cultivars with higher CRs and late FB germination had higher ABA content, lower GA$_3$ contents, and their enzyme activities were significantly higher than those of the cultivars with lower CR. Therefore, the changes in the content of various substances in FBs are related to the amount of cooling required by the cultivar and at what point the FBs germinate, and the changes in their contents can be used as one of the indicators for judging the dormancy process of FBs.

**Keywords:** *Prunus mume*; chilling requirement; physiological mechanism of dormancy

## 1. Introduction

*Prunus mume* Sieb. et Zucc. (Rosaceae), as a famous traditional Chinese flower, has very broad market appeal because of its elegant flower color, beautiful flower shape, and calming fragrance [1]. It is widely used in festivals and horticultural exhibitions. However, since *P. mume* generally blossoms in late winter and early spring, it cannot be shown in other periods. Therefore, adjusting the blossoming of *P. mume* to the target flowering period can not only meet the needs of the market and the public but also has important significance in terms of promoting the development of the flower industry.

At present, the most commonly used methods for flowering regulation include temperature regulation, light treatment, the external application of plant growth regulators,

and appropriate cultivation measures [2], and temperature is one of the leading factors for regulating the flowering stage of woody plants. *P. mume*, like most deciduous fruit trees, is very sensitive to temperature, so the regulation of its flowering period needs to be based on chilling requirements (CRs). CR refers to the number of hours of low temperature required for deciduous fruit trees to break natural dormancy. Generally, the increase in and exposure to chill is considered to be the most effective factors for plants to break dormancy [3]. Only when a certain low temperature level is reached can natural dormancy be successfully achieved. The successful completion of this process is an important stage that must be experienced in the subsequent growth and development cycle [4]. If chilling is not carried out, natural dormancy is not completed normally, even if the external conditions are suitable, which will inevitably cause growth and development disorders as flower buds (FBs) cannot germinate at the right time, or germinate irregularly, even causing deformity or the severe abortion of flower organs [5]. By mastering the CRs of cultivars, we can more accurately control the flowering period. At present, the most widely used estimation models for calculating the low temperature requirements for the dormancy of plants are the 7.2 °C model, the Utah model, and the dynamic model. Among these, the 7.2 °C model was proposed by Weinberger in the United States and is the simplest of many of the models [6,7]. The Utah model is the most widely used and considers the effect of low temperatures on growth and the offsetting effect of high temperatures on low temperatures [8]. The dynamic model considers both the positive effects of chilling and the negative effects of high temperatures on plant dormancy. However, this model has a disadvantage in that the method for estimating chilling demand is more suitable for tropical and subtropical regions [9]. Another downside of the dynamic model is that the calculation method is relatively complicated and has not yet been widely adopted [10–12].

As early as 1999, Ou Xikun used the "trial and error method" to conduct a computerized model study on the low temperature demand of Taiwan's *P. mume* fruit cultivars and evaluated the CRs of six local *P. mume* cultivars in Taiwan [13]. Zhuang Weibing et al. (2010) used the 7.2 °C model, the 0–7.2 °C model, and the Utah model to study the CRs of 75 *P. mume* cultivars in Nanjing [14]. The results showed that the CRs of the selected cultivars ranged from 183 CU–1090 CU and demonstrated that the Utah model is more suitable for the CR data of *P. mume* cultivars. Yang Yahui (2013) used the Utah model to obtain data on the CRs of "Meiren", "Meifan", and "Yanhong xingzhi" and found that the CRs were 425 CU, 471 CU, and 466 CU, respectively [15]. Zhihong Gao (2012) evaluated the chilling and heating requirements of six Japanese apricot species using three models and established the correlations between CRs, heating requirements, and flowering dates [16]. However, different species and different cultivars of the same species also have different CRs due to their own characteristics and external environments. Therefore, exploring the CRs of different cultivars is of great significance for the cultivation of *P. mume*.

During the process of the plant being exposed to low temperatures in order to release dormancy, a series of physiological changes occur in the FBs to resist the effects of low temperatures, which enable the plants to improve their resistance to external environmental changes, control FB dormancy, and provide a basis for the recovery of growth and development [17]. When the natural environment changes, the substances of osmotic adjustment in the FBs cooperate with each other to adjust the cell osmotic pressure and maintain the cell membrane structure [18]. The formation of the antioxidant system in plants also maintains the metabolic balance of the reactive oxygen species (ROS) to prevent the excessive accumulation of free radicals during dormancy and the subsequent poisoning of the plants. Therefore, the antioxidant enzyme system is closely related to the release of dormancy. In addition, plant endogenous hormones play a very important role in the process of inducing, maintaining, and releasing FB dormancy, and abscisic acid (ABA) participates in regulating the initiation and maintenance of dormancy [19]. Gibberellins, such as $GA_3$, may play a role in dormancy release [20]. In addition, a large number of studies have shown that the flowering of deciduous woody plants is not only affected

by a single type of hormone but also by the mutual promotion and mutual antagonism of hormones.

At present, the research on the CR and physiological mechanism of woody plants mainly focuses on ornamental peaches, pears, crabapple, peonies and other such plants, and there are few relevant studies on *P. mume*. In this research, four early flowering *P. mume* cultivars were used as experimental materials, which are widely found in the Henan Region, China. The Utah model was used to count the minimum CR of each cultivar and to explore the physiological changes in FBs during the artificial low-temperature dormancy release. This research aims to provide a theoretical basis for the regulation of the *P. mume* blossom's flowering period.

## 2. Materials and Methods

### 2.1. Plant Material

The experiment was carried out in the World Wintersweet Garden (34°09′ N, 114°06′ E) in Yanling, Henan, China, from September to December 2021. Eight-year-old grafted potted seedlings of *P. mume* 'Gulihong' (Figure 1a), 'Zaohualve' (Figure 1b), 'Nanjing gongfen' (Figure 1c), and 'Zaoyudie' (Figure 1d), four early flowering cultivars, were chosen. There were 30 plants of each variety, resulting in a total of 120 plants. The test seedlings were robust; compact; free of diseases and insect pests; with evenly distributed branches and plump and full buds; and were uniform in size, with a ground diameter of about 2.5 cm to 4.1 cm and a plant height of about 80 cm. The temperature controller of the cold storage was the XMK-7 system produced by the Zhejiang Yuyao Mingxing Refrigeration Parts Factory, and the temperature accuracy was 0.1 °C.

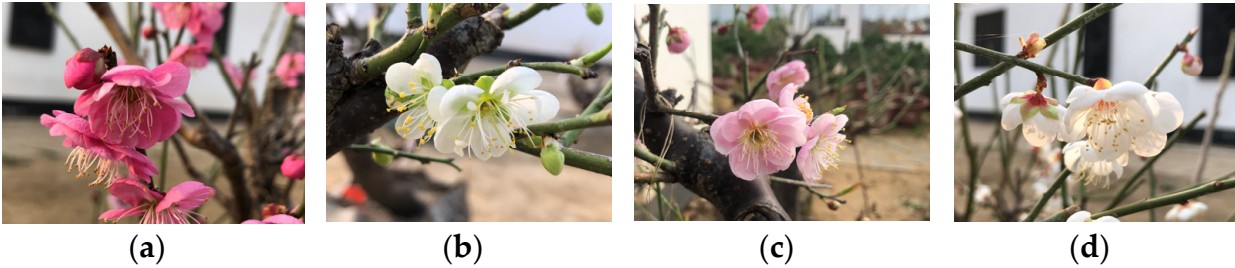

(**a**)　　　　　　　(**b**)　　　　　　　(**c**)　　　　　　　(**d**)

**Figure 1.** The *Prunus mume* cultivars selected for the experiment. Of these, (**a**) *P. mume* 'Gulihong' belongs to the Cinnabar Purple Form, with green branchlets, the reddish-brown skin of old branches, red inner skin, and deep red flowers. (**b**) *P. mume* 'Zaohualve' belongs to the Green Calyx Form; its calyx is green, the petals are saucer-shaped, and the flower color is white. (**c**) *P. mume* 'Nanjing gongfen' belongs to the Pink Double Form, with slanted branches and pink flowers. (**d**) *P. mume* 'Zaoyudie' belongs to the Alboplena Form, with milky yellow buds with red spots, and fronts of the flowers are yellow and white. These all belong to the Upright Mei Group, *Prunus mume* var. *typica*.

### 2.2. Experimental Method

#### 2.2.1. Method for Measuring Chilling Requirements

After the observation through a microscope of the internal differentiations of FBs in each cultivar was completed, the experimental materials were placed in constant temperature cold storage at 6 °C on September 6th for artificial low-temperature treatment, so that the dormant buds entered a deep dormancy state. A 2000 Lx lighting intensity and 8/16 h (light/dark) lighting conditions were used to simulate the average light intensity of this latitude in autumn, and water was regularly sprayed on the ground to maintain the relative air humidity of the experimental environment at 80%. In order to obtain more accurate CRs of different cultivars, three pots of each cultivar were removed from cold storage every two days from September 17th to 27th, and plants were moved out of cold storage six times in total. Thereafter, they were placed in a greenhouse, where routine management, such as regular watering and fertilization, was carried out. For each cultivar, three pots containing control plants were placed in a natural environment and underwent

routine cultivation management. In the experiment, the hourly temperature was recorded by a temperature and humidity recorder, combined with the meteorological data provided by the China Meteorological Data Network. After 30 days of cultivation from the day of sampling, the FB germination rates of different cultivars were counted, and the signs of releasing dormancy were quantified.

There were three pots for each cultivar, and three repeated experiments were set up. Three trunks on each pot of each cultivar were chosen, and each trunk contained around three flower branches (about 270 FBs). The germination was based on the cracking and germination (red dew) at the top of the FBs. After 30 days of cultivation from the day of sampling, the FB germination rates of different cultivars were observed and recorded every day to quantify the signs of releasing dormancy. The test utilizes the statistical method of Wang Lirong (1992) [21]. The date when the branch germination rate reaches 10% is the date that the plant is considered to be entering natural dormancy. When the germination rate of the FB gradually rises and stabilizes at 50% or above, the corresponding sampling date is regarded as the end of the natural dormancy of FBs; when the FB germination rate is between 60% and 70%, the average low temperature value during this sampling period and the previous sampling period is taken as the CR of the cultivar; and if the germination rate exceeds 70%, the low temperature level in the previous sampling period is directly used as the CR of the cultivar. The date of release from dormancy and the CRs level of different cultivars were determined by observing the average germination rate of FBs during low-temperature treatment. The experiment used the Utah model to calculate the CRs of four *P. mume* cultivars, and the unit used was the chill unit (CU). Its calculation formula is as follows:

$$Utah_t = \sum_{i=1}^{n} T_u \tag{1}$$

$$T_u = \begin{cases} 0, & T \leq 1.4\,°C \\ 0.5, & 1.4\,°C < T \leq 2.4\,°C \\ 1, & 2.4\,°C < T \leq 9.1\,°C \\ 0.5, & 9.1\,°C < T \leq 12.4\,°C \\ 0, & 12.4\,°C < T \leq 15.9\,°C \\ -0.5, & 15.9\,°C < T \leq 18.0\,°C \\ -1, & T > 18.0\,°C \end{cases} \tag{2}$$

where $Utah_t$ represents the cumulative value of cold temperature units at the end of dormancy, $T_u$ represents the actual low temperature per hour, and $T$ represents the recorded temperature per hour.

### 2.2.2. Method for Measuring Physiological Indicators in FB Dormancy Period

The above-mentioned four cultivars were used in the experiment, and nine pots of each cultivar were used. Plants with robust, well-developed growth that were healthy and free from insect pests and mechanical damage at the middle and upper parts of their crown peripheries were selected to determine the growth indicators and physiological indicators of FBs.

The FBs of the middle and upper parts of the annual shoots were selected one day before the low-temperature induction treatment (0 d) and every eight days after the start of the treatment. Each time the FBs were collected, the FBs were wrapped in tin foil and stored in liquid nitrogen, and then stored at −80 °C in an ultra-low temperature refrigerator for use in physiological index determination. A total of 0.5 g of fresh plant samples were weighed each time, and the soluble sugar (SS) content (anthrone colorimetric method, accurate to 0.01 mg·g$^{-1}$), starch (ST) content (anthrone colorimetric method, accurate to 0.01 mg·g$^{-1}$), superoxide dismutase (SOD) content (nitroblue tetrazolium photochemical reduction method, accurate to 0.01 U·g$^{-1}$·min$^{-1}$), peroxidase (POD) content (guaiacol method, accurate to 0.01 U·g$^{-1}$·min$^{-1}$), catalase (CAT) content (potassium permanganate titration method, accurate to 0.01 U·g$^{-1}$·min$^{-1}$), abscisic acid (ABA) content (enzyme-

linked immunosorbent assay ELISA, accurate to 0.01 ng·g$^{-1}$), Gibberellin (GA$_3$) content (ELISA, accurate to 0.01 ng·g$^{-1}$) were determined, following the methodology and experimental guidance of Li H [22] and Gao J [23]. The determination of each physiological and biochemical index was repeated 3 times per period.

*2.3. Statistical Analysis*

Statistical analysis was performed using Office 2019 for data processing, Origin Pro 2022 was used for drawing, and SPSS 26 software was used for principal component analysis and significance analysis.

## 3. Results

*3.1. Chilling Requirement for Dormancy Release*

The temperature data from the experiment are shown in Figure 2. The temperature statistics for the experiment were recorded from 12 September 2021 and concluded when all cultivars had reached their full flowering stage. The analysis of the daily maximum temperature, minimum high temperature, and hourly temperature indicated significant fluctuations. The average temperature during this period was 16.4 °C, with a relative humidity range of 30.2% to 89.3% and an average humidity of 58.2%. Notably, on 3 October, the temperature peaked at 33.4 °C, which was the maximum value within this period, followed by a fluctuating downward trend in temperature.

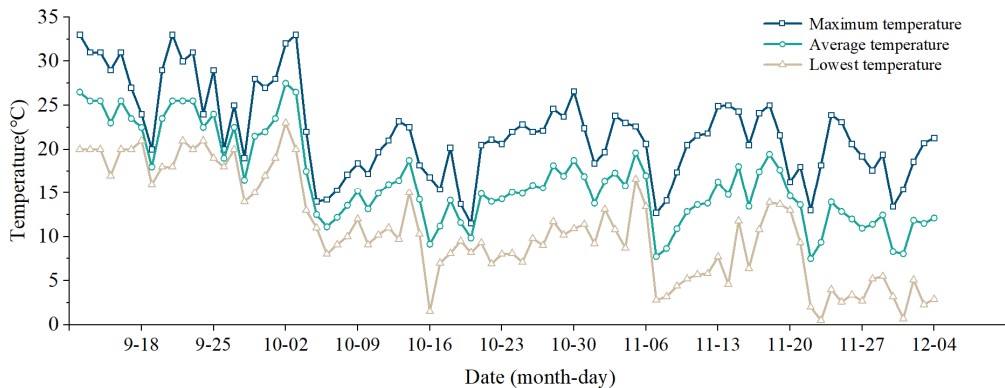

**Figure 2.** Changes of temperature in Yanling, Henan. The average temperature from 12 September to 4 December was 16.4 °C. The temperature peaked at 33.4 °C on 3 October, which was the maximum value within this period. The temperature during this period showed a trend of large fluctuations.

According to the sampling date corresponding to the FB germination rate, ≥50% is the end of dormancy, and the FB germination rate under the same low-temperature values was obtained via the experiments, as shown in Table 1. At the end of September, the FB germination rates of all cultivars reached more than 60%, and the release times of FB dormancy of the four cultivars was relatively close, most of which were concentrated in mid- and late September. It can be seen that 'Zaohualve' entered the dormancy period the earliest, on the 15th day after entering cold storage (21 September), and 'Gulihong' had the latest dormancy release time (24 September). However, none of the FBs of the control groups germinated during the entire course of the experiment.

Under the conditions of an artificial low temperature of 6 °C, the results among cultivars have certain differences. According to the Utah model, the low-temperature value of 'Gulihong' was the largest at 408 CU, and the low-temperature value of 'Zaohualve' was the smallest at 348 CU, followed by 'Nanjing gongfen' and 'Zaoyudie', which were 396 CU and 372 CU, respectively.

**Table 1.** Effects of different low-temperature values on the germination percentage rate of different *P. mume* cultivars.

| Release Date | CR (CU) | FB Germination Rate (%) | | | |
|---|---|---|---|---|---|
| | | 'Gulihong' | 'Nanjing Gongfen' | 'Zaoyudie' | 'Zaohualve' |
| 9-17 | 252 | 9.52 | 0 | 20.57 | 0 |
| 9-19 | 276 | 15.19 | 0 | 18.20 | 26.12 |
| 9-21 | 348 | 18.10 | 12.34 | 31.58 | 50.82 |
| 9-23 | 396 | 40.25 | 49.91 | 63.53 | 58.40 |
| 9-25 | 420 | 62.10 | 40.21 | 53.29 | 61.27 |
| 9-27 | 492 | 75.36 | 63.52 | 78.54 | 60.03 |

Note: CR, chilling requirement; CU, chill unit, which is the unit of the CR; FB, flower bud. The FB germination rate of each cultivar in the control group is 0.

Our results showed that FB germination rate was directly proportional to chilling accumulation and there was a positive correlation between refrigeration requirement and dormancy release date. However, without any artificial low-temperature treatment, the plants were still in a dormant state. This indicated that CR plays an important role in the flowering process of *P. mume*.

*3.2. Determination of Physiological Indicators during Low-Temperature Storage and Dormancy*

3.2.1. Changes of SS and ST Contents in FBs during Low-Temperature Release Dormancy

The changes of SS and ST contents in FBs of different cultivars during dormancy can be seen from Figure 3. Compared to the control group, which did not receive artificial low-temperature treatment, the contents of SS and ST in the FBs of different cultivars exhibited significant and regular changes when stored at 6 °C. However, the trend of content change in FBs of different cultivars is slightly different. The SS content of the four cultivars showed a fluctuating upward trend with the increase in the chilling level, while the ST content basically showed an opposite trend. In addition, before FB dormancy breaking, the SS content of 'Zaohualve', which has the smallest CR, was higher than in the other plants. When dormancy was completely released, the SS content of 'Gulihong', which required a higher amount of chilling, was significantly higher than that of the other cultivars. The ST content of 'Gulihong' was significantly higher than that of other cultivars at all stages.

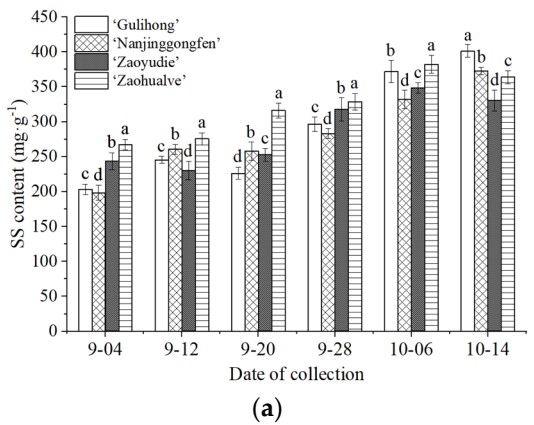

(a)

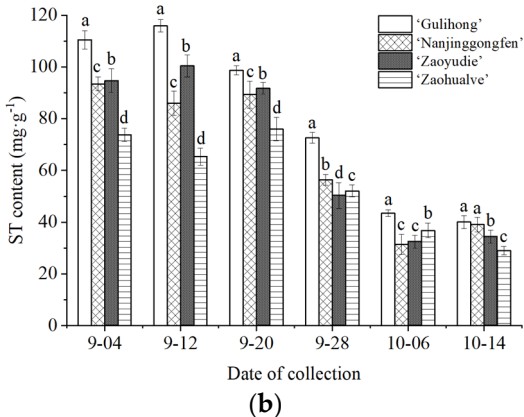

(b)

**Figure 3.** Changes in soluble sugar (**a**) and starch content (**b**) in the FBs of different *P. mume* cultivars during low temperature-induced dormancy. Notes: Error bars indicate the standard error, and different lowercase letters indicate significant differences at the 0.05 level among different cultivars in the same period (*p* < 0.05).

The results indicate that the low-temperature treatment triggered the stress response of the FBs, which resulted in changes in the substance content of the buds and promoted dormancy release. This also explains why the FBs in the control group, which did not

receive low-temperature treatment, failed to germinate. In addition, FBs can gradually accumulate SS during dormancy to provide the energy needed during dormancy and at the same time transform into ST to support germination and flowering during late dormancy. Cultivars with lower CR and early FB germination responded to low temperatures faster, and higher SS content was beneficial for releasing dormancy.

### 3.2.2. Effects of Low Temperatures on the Antioxidant Enzyme System in FBs during the Release of Dormancy

It can be seen from Figure 4 that the activities of various antioxidant enzymes in the FBs of different cultivars changed during dormancy. The dynamic change trends of enzyme activities in FB dormancy period, of each variety, are similar during the period of low-temperature dormancy release. The SOD activity was higher at the initial stage of low-temperature storage and then gradually decreased and remained at a low level. With the increase in cold, the POD activity showed an upward trend. The activity of CAT in the FB of each cultivar generally showed a trend of first increasing and then decreasing during the low-temperature storage. In addition, in the process of artificial low-temperature breaking dormancy release, the activity of antioxidant enzymes in the FBs of 'Gulihong', which requires a higher level of chilling, is higher than that of other cultivars, and the antioxidant enzyme activity of 'Zaohualve' is the lowest.

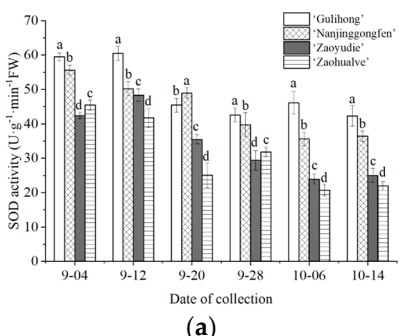
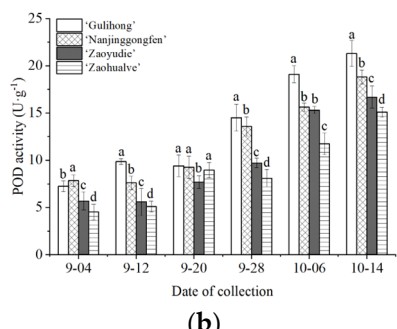
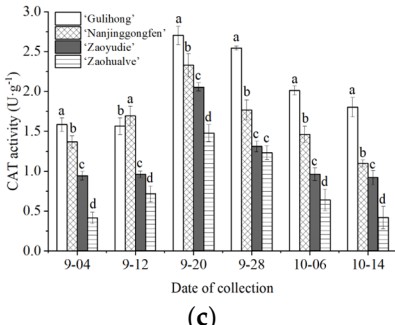

**Figure 4.** Changes of superoxide dismutase (**a**), peroxidase (**b**), and catalase content (**c**) in the FBs of different *P. mume* cultivars during low temperature-induced dormancy. Error bars indicate the standard error, and different lowercase letters indicate significant differences at the 0.05 level among different cultivars in the same period ($p < 0.05$).

The results indicate that, in contrast to the control group, which did not generate an active oxygen stress response, the enzyme activities of FBs from different cultivars showed variations during the artificial low-temperature treatment for dormancy release. The cultivars with higher CR demonstrated a stronger ability to scavenge free radicals and avoid the accumulation of reactive oxygen species in FBs.

### 3.2.3. The Effect of Low Temperatures on the Content Changes of Endogenous Hormones in FBs during the Release of Dormancy

The changes of ABA and $GA_3$ content in FBs of different cultivars during the period of dormancy can be seen from Figure 5. The content of $GA_3$ in the FBs of each cultivar was high before entering the cold storage, and after the initial low-temperature stress, the overall content showed a downward trend, while the ABA content gradually increased. When the dormancy was gradually released, the $GA_3$ content gradually increased and the ABA content decreased. In addition, during the process of breaking dormancy at low temperatures, the $GA_3$ content of cultivars with lower CRs was significantly higher than in others. The ABA content in FBs of 'Gulihong' with higher CR was higher than that of other cultivars with lower CR.

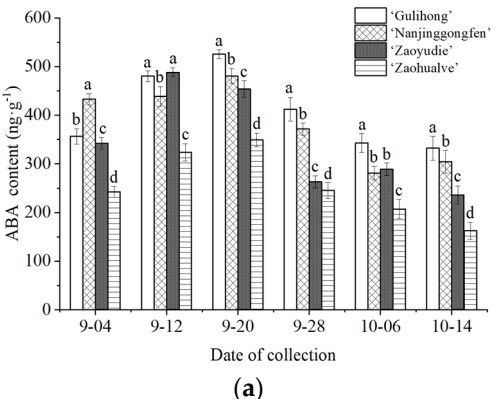

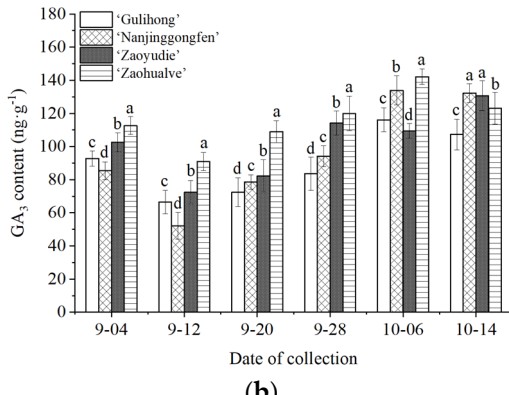

(a)    (b)

**Figure 5.** Changes of abscisic acid (**a**) and gibberellin (**b**) content in the FBs of different *P. mume* cultivars during low temperature-induced dormancy. Error bars indicate the standard error, and different lowercase letters indicate significant differences at the 0.05 level among different cultivars in the same period ($p < 0.05$).

Based on the experimental results, it can be concluded that low temperature caused changes in endogenous hormone levels during dormancy. In contrast, FBs in the control group that were not subjected to low-temperature treatment did not experience any stress responses, which resulted in the failure of the FBs to germinate. During the process of artificially inducing low temperatures to break dormancy, the FBs of cultivars that require less CR displayed lower levels of ABA content compared to other cultivars. This lower level of ABA content promoted the production of $GA_3$, which facilitated the earlier germination of FBs and ultimately resulted in earlier flowering.

## 4. Discussion

### 4.1. Chilling Requirement

Low temperatures are an important environmental factor that induce the initiation and release of dormancy in plant buds. Parkes (2020) produced a collection of CRs for eight apple cultivars (*Malus domestica*) in chill units from the autumn and winter of 2014 and 2015. The results showed that the range of CRs measured for Utah models was $976 \pm 40.3$ to $1307 \pm 86.6$ CU [24]. Li et al. (2019), based on the Utah model and the 7.2 °C model, determined that the CRs of *Chimonanthus praecox* (L.) Link (Calycanthaceae) cultivars "Suxin" and "Qingkou" were 558 CU and 570 CU, respectively, under natural conditions [25]. *P. mume* cultivars "Nanko" and "Ellching", from temperate Japan and subtropical Taiwan, required 500 and 300 chilling hours, respectively, to break FB dormancy [26]. In this research, through the artificial low-temperature treatment, the CRs of these cultivars were obtained. Of the cultivars included, 'Gulihong' had the highest CR of 408 CU, while 'Zaohualve' had the lowest CR of 348 CU. Through these data, we can infer that there are obvious differences in the CRs and dormancy states of different *P. mume* cultivars. The CR of each cultivar was different from "Mei Fan" [15], "Li Mei" [27], and others. It can be seen that different species and different cultivars of the same species have different CRs. It can thus be determined whether the plant blooms will be affected or not and when the flowering period will be. Moreover, the germination rate of each cultivar gradually increased with the increase in low-temperature treatment time, and the dormant state of FBs was effectively released after a certain period of treatment. However, without artificial low-temperature treatment, none of the FBs of the control groups germinated all the time. In this research, compared with the natural flowering period of *P. mume* blossoms from February to April, the flowering period of each cultivar was advanced to mid-November under artificial intervention. Therefore, CR is the main factor of the dormancy release. Mastering the level of CR can effectively control the flowering period.

*4.2. Carbohydrates*

During bud dormancy release, reserve carbohydrates play a critical role as they serve as the primary source of carbon and energy [28]. Moreover, SS and ST are known to function as signaling molecules in plant development regulation [29,30]. In order to resist the effects of low temperature, a large amount of carbohydrates are stored in plants to provide the basic substance required for dormancy consumption and sprouting growth [31]. This research found that, although the SS content in nectarine FBs in the early stage of dormancy was low, it showed a rising trend throughout the dormancy process and rose sharply in the late stage of dormancy; the change in ST content began to increase slowly in the early stage. After the accumulation reached its peak, it began to decline in the late dormancy period. The conclusion of this research is consistent with previous research results on Peach [32] and Peonia [33], which is further evidence that the effect of low temperature increases starch hydrolysis and, consequently, sugar content. The degradation of starch during natural dormancy is the reason for the increase in soluble sugar content, and the degradation of starch is related to the release of dormancy. Furthermore, prior to dormancy breaking, it was observed that the SS content of the cultivars requiring less CR was higher. This can be attributed to the fact that these cultivars respond more quickly to low-temperatures due to their lower CR. Moreover, a higher SS content is beneficial for the release of dormancy.

*4.3. Antioxidant Enzyme Activities*

In the period of the dormancy-releasing process, the activities of antioxidant enzymes change. Plants generate a higher level of oxidative stress at low temperatures, which leads to an increase in the level of reactive oxygen species (ROS) in the cells, thereby damaging the biological macromolecular structures, such as plant cell DNA, membrane lipids, and proteins. In order to protect cells from damage caused by oxidative stress, the activities of antioxidant enzymes in the plant ROS metabolic pathway are activated to counteract the effects of ROS [33]. Hernández (2021) studied the antioxidant metabolism during bud dormancy release in a low-chill peach variety. They found that the presence of $H_2O_2$-sensitive antioxidant enzymes in the floral buds could trigger oxidative signaling, leading to dormancy release [34]. The results of this research showed that the activity of SOD in FBs of each cultivar gradually decreased with the increase in cold, while the trend of POD activity was opposite to that of SOD activity, and the trend of CAT activity first increased and then decreased. Furthermore, the study also revealed that during the artificial low-temperature dormancy breaking process, the enzyme activity of 'Gulihong' was notably higher than that of the other cultivars. This suggests that cultivars with a higher CR have a stronger ability to scavenge free radicals and prevent the accumulation of reactive oxygen species in their FBs. Moreover, our results agree with a recent work on a high-chill peach variety [35]. It can be seen that different plant species have different changes in antioxidant enzyme activity during the dormancy period, which is different from the genome and metabolic pathways among tree species, which may lead to their different responses to environmental stress and oxidative stress during dormancy. Factors such as different environmental pressures, e.g., light and temperature, may also affect the regulation of enzyme activity, resulting in different enzyme activity changes in the FBs of different plants [36]. Therefore, more studies are needed to understand the enzyme activity changes in FBs of different tree species during dormancy to further understand the molecular mechanism of plant dormancy.

*4.4. Endogenous Hormone*

Endogenous hormones also play an important role in the process of inducing and maintaining FB dormancy in plants [17]. ABA is a type of plant hormone that has been widely recognized as a powerful growth inhibitor, while $GA_3$ is known to act as an antagonist to ABA, which in turn promotes the process of germination [37]. Two core hormones regulate bud dormancy status antagonistically [38]. In *P. mume*, the ABA/GA ratio was

reported to steadily decline during the dormancy release process [39,40]. In this research, during the low-temperature release period, the ABA content increased and reached the peak before the release of dormancy, and its content showed a downward trend with the passage of time and the increase in cold. The change of $GA_3$ content was the opposite. After 32 days of low-temperature storage, its content reached a maximum and, on the whole, remained at a relatively high level, indicating that $GA_3$ is the main hormone-like factor that promotes the release of dormancy and is important for FB germination and dormancy induction. It has a certain promoting effect. In addition, in the process of breaking dormancy at artificial low temperatures, 'Gulihong', with higher CRs, had higher ABA content and lower $GA_3$, while the cultivars with lower CRs were the opposite. These results agree with the conclusions of Wen [39]. It can be seen that cultivars with lower CRs have lower ABA concentrations and higher $GA_3$ concentrations, which resulted in early bud burst. High CRs and the late germination of FBs may be related to high ABA content and low IAA content.

## 5. Conclusions

These research results are of significance for *P. mume* cultivation. In this research, based on the method of artificial low temperatures used to release dormancy, by artificially maintaining low-temperature conditions of 6 °C, the CRs of 'Gulihong', 'Nanjing gongfen', 'Zaoyudie', and 'Zaohualve' were 408 CU, 396 CU, 372 CU, and 348 CU, respectively. This research has broad application value in horticulture and agriculture. In horticulture, based on the use of artificial low temperature to release dormancy, as performed in this study, mastering the level of chilling required by different cultivars can determine when to take chilling measurements, according to the specific showing time required, thus providing a method for accurately controlling the supply of flowers at times of major festivals and major horticultural exhibitions. This study achieves the early flowering of *P. mume*, which will help to increase yield and promote the development of the *P. mume* industry.

In addition, this research found that the process of releasing dormancy at low temperatures is closely related to the content of osmotic regulators, antioxidant enzyme systems, and endogenous hormones in FBs. The contents of different cultivars that require chilling are also different. We can use the change of contents as an indicator to judge the dormancy process. By measuring the growth and physiological indicators, this study was able to further explore the physiological mechanism of FBs during the period of low-temperature dormancy release, as well as the internal relationship between the level of chilling required, the time of FB germination, and the physiological changes in the FB, and provide a basis for forced cultivation methods of *P. mume*.

In the future, the dormancy mechanism of *P. mume* with different CRs can be further analyzed at the molecular levels, such as transcriptome and metabolome, so as to provide a more scientific basis for *P. mume* cultivation.

**Author Contributions:** Conceptualization, Q.L. and K.M.; methodology, Y.Z.; software, Y.Z.; validation, Y.Z.; formal analysis, Y.Z.; investigation, Y.Z.; resources, Q.L. and K.M.; data curation, Y.Z.; writing—original draft preparation, Y.Z.; writing—review and editing, Y.Z.; visualization, Y.Z.; supervision, Q.L. and K.M.; project administration, K.M.; funding acquisition, Q.L. and K.M. All authors have read and agreed to the published version of the manuscript.

**Funding:** This research was funded by National Key Research and Development Program (2020YFD1000500) sub-project "Integration and Demonstration of Light, Simple and Efficient Cultivation Technology for *Prunus mume*" (2020YFD100050201).

**Data Availability Statement:** No new data were created or analyzed in this study. Data sharing is not applicable to this article.

**Conflicts of Interest:** The authors declare no conflict of interest. The funders had no role in the design of the study; in the collection, analyses, or interpretation of data; in the writing of the manuscript; or in the decision to publish the results.

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
