# Peer review of "Research on Chilling Requirements and Physiological Mechanisms of Prunus mume"

_horticulturae, doi:10.3390/horticulturae9050603_

Round 1
Reviewer 1 Report
The topic is interesting and relevant; though, the article must be improved (see file attached).
Although the journal is fexible about the order of the different sections of the articles to be published, in this case, I consider important to first present the Methods and afterwards, the Results.
Introduction: congruent with the objectives of the research.
Materials and Methods: they are weak, not well explained or not explained at all; therefore, methods are confused and leading to a lot of gaps, and these gaps are reflected in the Results. It is not clear how many samples were analyzed in total (8 pots/4 cultivars; hence, how many FBs/plant/cultivar were used, etc.); hence, are these samples replcates or subreplicates? Authors mention "controls" for supporting the experimental research; however, they never mention how many samples/cultivar were used as controls nor how they were experimentally treated, neither any result obtained. It is also lacking the statistical analysis. In addition, it is not clear if the cultivars are taxonomic/genetic varieties (it is clear they belong to the same species), different ecotypes (living naturally in different habitats -need to describe and compare them-) or if they were the result of human manipulation and if it is possible to consider them as biologically different. It looks like that the cultivars are different ecotypes; thus, it is expected to be obtained different results, due to the degree of adaptation of each ecotype to its specific habitat. This section really needs to be improved.
Results: questionable because of the weakness of the Material and Methods section. They are understandable but lacking certainty. Tables and Figures also need to be improved (a title and more information in the legend).
Discussion: adding the weakness of the Material and Methods, and Results, sections, it is also weak because it is not well supported; even though, it is understandable.
Conclusion: though it is quite general, it is congruent.
References: congruent.

Author Response
Thank you very much for your valuable comments and suggestions. Based on your comments and suggestions, I have finished revisions and the response to the your comments.The underlined text is the modified content in the text. Please see the attachment. To facilitate your review, the revised parts have been marked in red in the revised manuscript. Thank you again for your valuable comments and suggestions.

Reviewer 2 Report
Article
Research on chilling requirement and physiological mechanism of Prunus mume
Yuhan et al., 2023
General comments:
- It’s a good idea a summarize the main conclusions.
- Why is important your research? How can you apply the production of this knowledge in the horticulture and agriculture?
Line 35: Change mum by mume.
Line 57: When you write by first time flower bud as FB, you have to write flower bud (FB). I read in the abstract that you have of this form, but in the introduction you have to write the text and then the abbreviation in brackets.
Line 65-68: Rewrite this paragraph more clearly.
Line 70: You write CRs, but nowhere in the text does it specify that it is. If this is “chilling requirement” you have to specific this abbreviation in brackets.
Line 202, 216, 242, 259: Put P of plant in capital letter.
Line 296-300: How do you measure soluble sugars, starch, SOD activity, POD activity, CAT activity, ABA and GA3? This is an important part of the methodology and is omitted. You must write of this in materials and methods.
Line 312: put in italic letter the scientific name P. mume. After the P. letter, you must put space.

Author Response

(The authors gave the same response as above.)

Round 2
Reviewer 1 Report
The authors have really improved the manuscript, mainly in the Methods.
Some comments for improving, even more, the article:
i) It is still important to improve the title and the legends of Figures and Tables, which ought to be understandable by themselves, without reading any part of the article.
ii) Nothing is told about the Controls in the Results section. Controls have a lot of information to give, even if they did not react at all. Life sciences is about comparing (mainly in experiments) among characters, species, habitats, etc., and the frame is always the Control. There is a sentence about them in the Discussion.
Few comments were done in the PDF file.
The article is scientifically relevant.

Author Response
Thank you very much for your valuable comments and suggestions. According to your comments and suggestions in the second round, this manuscript has been revised and explained, please see the attachment.

Reviewer 2 Report
Dear Authors,
I'm very grateful for the corrections, but there is still a lack of information in the methodology that you must add. Please work in this.
Line 114, 142, 143, 145, 179... : if a number is <10 you must write it in letters. From 10 you must write as numbers. Please check all document.
Line 137, 143: "September 6" change by "September 6th"; September 17 by "September 17th"; "September 27" by September 27th.
Line 153: Figure 2 is a part of the result. I think that is better put in context in the results paragraphs.
Line 155 and 244: These figures have the same number
Line 186-196: Where is the references of the methodology? This section should describe in detail each one of the methodology and the respective references of each experiment.
You correct it some parts, but you have to explain the methodology or write the references (i.e.: anthrone colorimetric method. reference? steps?)

Author Response

(The authors gave the same response as above.)

Round 3
Reviewer 1 Report
The article really improved with these last corrections.